# Montmorillonite-Based Natural Adsorbent from Colombia for the Removal of Organic Pollutants from Water: Isotherms, Kinetics, Nature of Pollutants, and Matrix Effects

**Marcela Paredes-Laverde** [1,2]**, Diego F. Montaño** [3] **and Ricardo A. Torres-Palma** [1,*]

1 Grupo de Investigación en Remediación Ambiental y Biocatálisis (GIRAB), Instituto de Química, Facultad de Ciencias Exactas y Naturales, Universidad de Antioquia UdeA, Calle 70 No. 52-21, Medellín 50011, Colombia

2 Grupo de Investigación Cuidados de la Salud e Imágenes Diagnósticas, Facultad de Ciencias de la Salud, Fundación Universitaria Navarra—Uninavarra, Calle 10 No. 6-41, Neiva 31008, Colombia

3 Grupo de Investigación Ciencia y Diseño de Materiales (CiDiMat), Departamento de Química, Facultad de Ciencias Básicas, Universidad de Pamplona, Km 1 Via Bucaramanga Ciudad Universitaria, Calle 5 No. 3-93, Pamplona 26909, Colombia

* Correspondence: ricardo.torres@udea.edu.co; Tel.: +57-315-314-98-76

**Abstract:** The presence of dyes and pharmaceuticals in natural waters is a growing concern worldwide. To address this issue, the potential of montmorillonite (MMT), an abundant clay in Colombia, was assessed for the first time for the removal of various dyes (indigo carmine (IC), congo red (CR), methylene blue (MB) and crystal violet (CV)) and pharmaceuticals (levofloxacin and diclofenac) from water. Initially, the MMT was characterized. TGA and FTIR showed OH groups and water adsorbed onto MMT. XRD showed an interlayer spacing of 11.09 Å and a BET surface area of 82.5 m$^2$g$^{-1}$. SEM/EDS revealed a typical flake surface composed mainly of Si and O. Subsequently, the adsorbent capacity of MMT was evaluated for the removal of the pollutants. Adsorption isotherms showed a fit to the Langmuir model, which was confirmed by the Redlich–Peterson isotherm, indicating a monolayer-type adsorption. Furthermore, adsorption kinetics were best described by the pseudo-second-order model. Adsorption capacity (for dyes CV > MB > CR > IC) was associated with the attractive forces between the contaminants and MMT (PZC 2.6). Moreover, the findings evidenced that MMT can remove MB, CR, CV, and levofloxacin by electrostatic attractions and hydrogen bonding, while for IC and diclofenac only hydrogen bonding takes place. It was shown that MMT was most cost-effective at removing CV. Additionally, the material was able to be reused. Finally, the MMT efficiently removed CV in textile wastewater and levofloxacin in urine due to the positive charge of the pollutants and the low PZC of MMT.

**Keywords:** pharmaceuticals; dyes; clays; adsorption; isotherm; kinetics; water treatment





## 1. Introduction

Textile industries consume large quantities of water in dyeing processes, generating considerable amounts of colored wastewater with anionic and cationic dyes. Both types of dyes in water, even at low concentration levels, can cause dramatic decreases in light transmittance, thus greatly affecting the photosynthesis of aquatic plants and seriously damaging the ecological environment [1]. In fact, the presence of anionic and cationic dyes in rivers, lakes, ponds, and wastewater treatment plant effluents have been reported at concentrations at or above 0.01 μg L$^{-1}$ [2], while in textile wastewaters, the presence of anionic and cationic dyes can reach up to 45,000 μg L$^{-1}$ [3].

The presence of pharmaceuticals in water resources is also of great concern. Pharmaceuticals are stable compounds created to improve human health and promote human well-being. There have been increasing concerns about the presence of pharmaceuticals in the environment due to the negative environmental impact of such compounds, such as

their associated toxicity and their main role in the proliferation of antibacterial resistance, even at very low concentrations [4]. Concentrations of pharmaceuticals from 0.006 µg L$^{-1}$ to 50 µg L$^{-1}$ have been reported in the influent and effluent of wastewaters [5], while in river water it is possible to find concentrations in the range of 8.62–590.6 µg L$^{-1}$ [6]. In aqueous matrices such as urine, concentrations of pharmaceuticals from 1 µg L$^{-1}$ to 6400 µg L$^{-1}$ have been reported [7].

According to the literature, the most cost-effective alternatives to deal with the presence of organic pollutants in waters include membrane separation, flocculation, ion exchange, and adsorption [8]. Adsorption systems using clays present numerous advantages, such as a relatively short time of operation, easy application, and a cheap cost. In addition, clays have stone properties, are resistant to aggressive media, do not require additional purification after secondary use, can be employed in large quantities, and contain highly dispersed hydroaluminosilicates, which have proven potential to effectively remove a variety of pollutants [9].

Clays such as bentonite, kaolin, and montmorillonite have been shown to be effective at the removal of dyes and pharmaceuticals [8]. Montmorillonite (MMT) is of special interest because it is one of the main components of clay minerals. MMT's structure is that of an octahedron and tetrahedron (TOT), with two layers of tetrahedral silica sheets sandwiching one octahedral aluminum sheet, where the interlayer charge can be neutralized by the presence of cations such as sodium or calcium [10]. Additionally, MMT has a specific surface area of around 74.2 m$^2$ g$^{-1}$, which is much higher than the surface areas of other clays such as illite (11.2 m$^2$ g$^{-1}$) and kaolinite (15.3 m$^2$ g$^{-1}$) [11]. Thus, MMT could be a key inorganic adsorbent for the adsorption of pollutants from water. However, in spite of the relative abundance of MMT in Colombia, few reports have evaluated the ability of MMT from Colombia to remove a variety of organic pollutants, particularly dyes and pharmaceuticals, from water.

Therefore, the objective of this study is to evaluate, for the first time, the use of MMT from Colombia as a versatile natural adsorbent to remove pollutants with different natures and chemical structures from water. Common anionic dyes—such as indigo carmine (IC) and congo red (CR)—and cationic dyes, such as methylene blue (MB) and crystal violet (CV)—were selected for this study. In addition, two highly consumed pharmaceuticals—the antibiotic levofloxacin (zwitterionic structure) and the analgesic diclofenac (anionic structure)—were considered. For these purposes, MMT from Bugalagrande, Colombia, was obtained and properly characterized. Subsequently, the ability of MMT to remove IC, CR, MB, and CV from distilled water was tested. The effects of the adsorbent dose and the type of pollutant were then addressed. To better understand the results, the experimental data were adjusted to Langmuir, Freundlich, and Redlich—Peterson adsorption isotherms, and pseudo-first order and pseudo-second order kinetics models were assessed. Moreover, the most probable interactions of dyes containing MMT were proposed using FTIR analysis before and after the adsorption process. Subsequently, the economic costs of the process and the reuse of the MMT were studied. Finally, to evaluate the effects of a matrix using MMT, the dye with the highest percentage of adsorption in distilled water was studied in textile wastewater. To further validate the potential and versatility of the technology, the adsorption of the pharmaceuticals levofloxacin (LEV) and diclofenac (DCF) in both distilled water and urine (a very complex matrix) was investigated.

## 2. Materials and Methods

### 2.1. Reagents

Congo red (CR) ($C_{32}H_{22}N_6Na_2O_6S_2$), indigo carmine (IC) ($C_{16}H_8N_2Na_2O_8S_2$), methylene blue (MB) ($C_{16}H_{18}ClN_3S$), crystal violet (CV) ($C_{24}H_{28}N_3Cl$), levofloxacin (LEV) ($C_{18}H_{20}FN_3O_4$), diclofenac (DCF) ($C_{14}H_{11}Cl_2NO_2$), formic acid ($CH_2O_2$), acetonitrile ($C_2H_3N$), sodium chloride (NaCl), urea ($CH_4N_2O$), sodium dihydrogen phosphate ($NaH_2PO_4$), sodium sulfate ($Na_2SO_4$), ammonium chloride ($NH_4Cl$), potassium chloride (KCl), magnesium chloride six hydrate ($MgCl_2 \bullet 6H_2O$), calcium chloride two hydrate

(CaCl$_2$•2H$_2$O), and sodium hydroxide (NaOH), all of analytical grade, were supplied by Merck S. A. (Darmstadt, Germany). Other chemical reagents, such as sodium carbonate (Na$_2$CO$_3$), sodium bicarbonate (NaHCO$_3$), sulfuric acid (H$_2$SO$_4$), and starch were provided by Sigma Aldrich (St. Louis, MO, USA). All solutions were prepared in distilled water. Additionally, the montmorillonite clay (MMT) with a general chemical formula of (Si$_4$)$^{IV}$(Al$_4$)$^{VI}$O$_{10}$(OH)$_8$ used in this study was obtained from Bentocol (Bugalagrande, Valle del Cauca, Colombia).

### 2.2. MMT Preparation

50 g of MMT was added to 500 mL of distilled water to eliminate the quartz fraction. It was then vigorously stirred for 2 h in a Labscient ultrasound instrument (power 360 W), after which the aqueous suspension was vacuum filtered using a Roker pump (power 125 W) for 15 min, thereby generating a solid. Later, the clay was dried at 110 °C for 12 h in a drying oven from Memmert (power 1700 W) and passed through a sieve with a particle size < 200 μm. Finally, 45 g of MMT was obtained, which was stored in a glass container for its posterior use as the adsorbent.

### 2.3. Characterization of MMT

The MMT was characterized by thermogravimetric analysis (TGA), Fourier transform infrared spectroscopy (FTIR) and scanning electronic microscopy (SEM). The TGA spectra was obtained using a Q500 thermogravimetric analyzer from TA Instruments (New Castle, DE, USA), and the heating rate was 10 °C min$^{-1}$ from room temperature up to 800 °C in air. The FTIR was recorded in the range of 4000–400 cm$^{-1}$ using attenuated total reflectance—ATR (PerkinElmer, Waltham, MA, USA). SEM analysis was carried out using a FEI Quanta 250 microscope (OR, USA), coupled with an energy dispersive spectrometer (EDS) to identify the relative content of elements.

An X-ray diffraction (XRD) experiment was performed using an XPert PANalytical Empyrean diffractometer (EA Almelo, The Netherlands) at 40 mA and 40 kV with monochromatized Cu K$\alpha$ radiation (λ of 1.540598 Å and 2θ = 0–60°). Furthermore, the interlayer spacing of the clay was determined through the integration of the peaks reported by the MMT in the XRD spectrum. Additionally, the point of zero charge (PZC) of the MMT was calculated using the solid addition method [12]. Finally, the specific surface area was determined by nitrogen physisorption applying the Brunauer-Emmett-Teller (BET) equation to the isotherm [13] using a Micrometrics ASAP 2020 analyzer.

### 2.4. Adsorption Experiments

Initially, solutions at concentrations of 5.74 mg L$^{-1}$ of IC, 8.57 mg L$^{-1}$ of CR, 3.93 mg L$^{-1}$ of MB and 4.85 mg L$^{-1}$ of CV, which represent a dye concentration of 1.23 × 10$^{-2}$ mmol L$^{-1}$, were prepared. 100 mL of each solution was placed in contact with 0.02 g of MMT and stirred at 200 rpm. Samples were taken at different intervals of time for 120 min. Subsequently, the adsorbed amounts of IC, CR, MB and CV were calculated using Equation (1) [12]. Likewise, to understand the effects of MMT dose and pollutant structure (anionic and cationic dyes), all dyes (at 1.23 × 10$^{-2}$ mmol L$^{-1}$) were evaluated using adsorbent doses of 0.2, 0.6, 1.0 and 2.0 g L$^{-1}$. The amount of dye adsorbed onto MMT was then calculated using Equation (1). All the experiments were carried out at the natural pH of the solutions (IC: 5.7; CR: 5.7; MB: 5.6; CV: 6.0). Once these experiments were performed, the dose of MMT that was best at removing the pollutants was selected and the effect of the matrices was evaluated. Aqueous matrices such as textile wastewater [14] and urine [15] were prepared in the laboratory in accordance with previous reports (See Supplementary Materials Table S1). Both matrices were doped with a concentration of 1.23 × 10$^{-2}$ mmol L$^{-1}$ of contaminant (in the order of mg L$^{-1}$). The textile wastewater was doped with 4.85 mg L$^{-1}$ of CV, while the urine was doped with 4.44 mg L$^{-1}$ of LEV or with 3.64 mg L$^{-1}$ of DCF. Consequently, 100 mL of each of the doped matrices under continuous stirring were put in contact with 0.1 g of MMT, and samples were taken at different time intervals over a 60 min period. The same

procedure was performed using distilled water as a control test. The removal of pollutants was determined by calculating the adsorption percentage (%) using Equation (2) [12].

$$Adsorbed \ Amount = \frac{C_0 - C}{m}V \tag{1}$$

$$Adsorption \ (\%) = \frac{C_0 - C}{C_0} * 100 \tag{2}$$

where $C_0$ and $C$ (mg L$^{-1}$) are the liquid phase concentrations of the pollutant at initial and equilibrium time, respectively, $V$ (L) is the volume of the solution and $m$ (g) is the mass of MMT. In some cases, the *Adsorbed Amount* (mg g$^{-1}$) was converted to mmol g$^{-1}$ using the molecular weight of the pollutants.

To determine the cost of the process, the commercial value of 1 kg of MMT, and the energy and water required in the preparation process were considered. Thus, the cost of using MMT for the removal of dyes was determined from the ratio of the total production cost of the clay with a $q_m$ of CV, IC, MB and CR.

To evaluate the reuse potential of the material, the MMT used was put in contact with 20 mL of HNO$_3$ (1 M) and 20 mL of HCl (1%) in ethanol for a period of 5 h. Subsequently, the material was washed with distilled water and dried at 105 °C for 24 h. After this time, the material was stored to be used in the next adsorption cycle.

*2.5. Equilibrium Isotherms*

The removal kinetics of $1.23 \times 10^{-2}$ mmol L$^{-1}$ of IC (5.74 mg L$^{-1}$), CR (8.57 mg L$^{-1}$), MB (3.93 mg L$^{-1}$), and CV (4.85 mg L$^{-1}$) in distilled water using 0.2–2.0 g L$^{-1}$ of MMT were adjusted to the Langmuir, Freundlich and Redlich-Peterson isotherms. The Langmuir isotherm assumes monolayer sorption onto a surface containing a finite number of sorption sites of uniform sorption strategies with no transmigration of sorbate in the surface plane [1]. The Langmuir isotherm was calculated using the following equation (Equation (3))[9]:

$$q_e = \frac{q_m K_L C_e}{1 + K_L C_e} \tag{3}$$

Equation (3) can be rearranged into a linear form:

$$\frac{C_e}{q_e} = \frac{C_e}{q_m} + \frac{1}{K_L q_m} \tag{4}$$

where $C_e$ (mg L$^{-1}$) and $q_e$ (mg g$^{-1}$) are the equilibrium concentration and the amount of adsorbate adsorbed per unit mass of adsorbent, respectively, and $q_m$ (mg g$^{-1}$ or mmol g$^{-1}$) is the amount of dye adsorbed per unit mass of adsorbent equivalent to the formation of a complete monolayer. The affinity constant, $K_L$ (L mg$^{-1}$), is the equilibrium constant of the adsorption process. Thus, from a plot of $C_e/q_e$ vs. $C_e$, $q_m$ can be obtained from the slope and $K_L$ from the intercept. The essential characteristics of the Langmuir isotherm can be expressed by a dimensionless equilibrium parameter (known as the separation factor), which is defined as follows:

$$R_L = \frac{1}{1 + K_L C_o} \tag{5}$$

where the $R_L$ value indicates the type of adsorption: irreversible ($R_L = 0$), favorable ($0 < R_L < 1$), linear ($R_L = 1$) or unfavorable ($R_L > 1$); and $C_0$ (mg L$^{-1}$) is the initial concentration of the metal.

The Freundlich model is an empirical equation based on sorption on heterogeneous surfaces or surfaces supporting sites of varied affinities. This was studied using Equation (6) [16].

$$q_e = K_F C_e^{1/n} \tag{6}$$

where $K_F$ and $n$ are indicators of the adsorption capacity and the adsorption intensity, respectively. If the $n$ value lies in the range between 1 and 10, favorable adsorption takes place [12]. Rearranging Equation (6) gives the following:

$$ln q_e = ln\, K_F + \frac{1}{n} ln\, C_e \tag{7}$$

The Redlich–Peterson equation is widely used as a compromise between the Langmuir and Freundlich systems. This model has three parameters. It incorporates the advantageous significance of both models and can be represented as follows in Equation (8) [17].

$$q_e = \frac{K_{RP} C_e}{1 + a_R C_e^\beta} \tag{8}$$

where $K_{RP}$ $\left(L\,g^{-1}\right)$ and $a_R$ $\left(L\,mg^{-1}\right)^\beta$ are the Redlich-Peterson isotherm constants and $\beta$ is an exponent that lies between 0 and 1. Equation (8) can be linearized by taking logarithms of both the sides, as shown in Equation (9):

$$ln\left(K_{RP}\frac{C_e}{q_e} - 1\right) = \ln a_R + \beta \ln C_e \tag{9}$$

*2.6. Adsorption Kinetics*

Pseudo-first order and pseudo-second order kinetic models were calculated from the results obtained when removing $1.23 \times 10^{-2}$ mmol $L^{-1}$ of IC (5.74 mg $L^{-1}$), CR (8.57 mg $L^{-1}$), MB (3.93 mg $L^{-1}$), and CV (4.85 mg $L^{-1}$) from distilled water with a dose variation between 0.2–2.0 g $L^{-1}$ of MMT.

The pseudo-first order kinetic model assumes that one active sorption site on the clay is adsorbing one pollutant molecule [1]. Using this model, the rates of adsorption of IC, CR, MB and CV on MMT were determined by the following equation [12]:

$$ln\,(q_e - q_t) = ln\, q_e - k_1 t \tag{10}$$

where, $q_t$ is the amount of dye adsorbed (mg $g^{-1}$) at the time $t$ (min), $q_e$ is the amount of pollutant adsorbed at the equilibrium (mg $g^{-1}$), and $k_1$ is the equilibrium rate constant of pseudo-first order adsorption (min$^{-1}$). Thus, from Equation (10), $ln\,(q_e - q_t)$ vs. $t$ was plotted, obtaining $q_e$ from the intercept and $k_1$ from the slope.

The pseudo-second order equation assumes that two active sorption sites on the MMT are adsorbing one dye molecule [1]. The pseudo-second order model can be expressed in the following form [17]:

$$\frac{t}{q_t} = \frac{1}{k_2 q_e^2} + \frac{t}{q_e} \tag{11}$$

where $k_2$ is the pseudo-second order model rate constant of adsorption (mg $g^{-1}$ min$^{-1}$). Thus, the plots of $t/q_t$ vs. $t$, allow $q_e$ and $k_2$ to be determined from the slope and the intercept, respectively.

*2.7. Evaluation of Applicability of the Isotherm Models and Kinetics*

The comparisons of the applicability of each isotherm model and kinetics were based on the linear coefficient correlation ($R^2$), the average percentage error ($APE$) (Equation (12)) and the normalized standard deviation ($\Delta q$, %) (Equation (13)) [18].

$$APE\,(\%) = \frac{\sum_{i=1}^{N}\left|\left(\frac{q_{exp} - q_{cal}}{q_{exp}}\right)\right|}{N} * 100 \tag{12}$$

$$\Delta q \; (\%) = 100 \; \sqrt{\frac{\Sigma \left( \frac{q_{exp} - q_{cal}}{q_{exp}} \right)^2}{N - 1}} \tag{13}$$

where $q_{exp}$ and $q_{cal}$ are the experimental and calculated amounts of dye adsorbed at equilibrium, respectively, and $N$ the number of measurements. As can be seen, the $APE$ and $\Delta q$ indicates the fit between the experimental and predicted values of the adsorbed amounts.

### 2.8. Analytical Techniques

The quantitative determination of IC, CR, MB and CV were done by UV visible spectroscopy in a UV5 Mettler Toledo spectrophotometer set at 611 nm [19], 498 nm, 661 nm and 589.5 nm [20], respectively. Pharmaceutical evolutions were observed by liquid chromatography using a Thermo Scientific UHPLC (Dionex Ultimate 3000) equipped with a diode array detector (DAD) and an Acclaim™ 120 RP C18 column (5 μm, 4.6 × 150 mm). LEV removal was monitored at 290 nm, using mobile phase formic acid/acetonitrile 85/15 (% $v/v$) and a flow of 1.0 mL min$^{-1}$ in isocratic mode. In addition, DCF removal was observed at 260 nm, with a mobile phase of formic acid/acetonitrile 30/70 (% $v/v$) and flow isocratic mode of 0.5 mL min$^{-1}$. In all cases, the injection volume was 25 μL. All the experiments were conducted in at least duplicate.

## 3. Results and Discussion

### 3.1. MMT Characterization

The composition and thermal stability of the MMT was studied by TGA (Figure 1a). The TGA showed three events. In the first and second event, the weight losses below 250 °C can be attributed to the moisture and water from the interlayer spacing of the clay, respectively. The subsequent event occurred gradually between 250 °C and 700 °C, which can be correlated with dehydroxylation of MMT [21].

In addition, the presence of water in the interlayer spacing and hydroxyl groups in the MMT was confirmed by FTIR analysis (Figure 1b). A band at 3698 cm$^{-1}$ showed the presence of Si-OH, while the band at 3620 cm$^{-1}$ was associated with the Al-OH group [22]. In turn, the bands around 3425 cm$^{-1}$ and 1635 cm$^{-1}$ are assigned to the stretching and bending of the hydroxyl group in the absorbed water, respectively. Likewise, bands related with the Si-O bonds were observed at 1038 cm$^{-1}$ for Si-O-Si, 520 cm$^{-1}$ for Si-O-Al, and 463 cm$^{-1}$ for Si-O-Fe [23]. Other bands at 910 cm$^{-1}$ and 820 cm$^{-1}$ suggest the presence of Al-Al-OH and Al-Mg-OH, respectively in the clay [24]. This series of chemical compounds present in the MMT was confirmed by EDS analysis (Table 1). The MMT showed a high content of O, Si, Al and Fe. Other elements such as C, N, Mg and Na were also reported.

On the other hand, the crystallographic structure of MMT was determined by XRD (Figure 1c). The XRD pattern of the clay shows different peaks at 2 theta: 7.97, 20.08, 28.34, 35.99, 58.59, which were accordingly indexed to the (001), (100), (005), (110) and (210) diffractions of MMT [23]. In this study, 7.97 (2 theta) is a typical diffraction peak of MMT corresponding to an interlayer spacing (d001) of 11.09 Å. In fact, the interlayer spacing found in this study was greater than for other single (5.278 Å [25], 9.8 Å, 10.3 Å [26]) and modified (6.5 Å [27]) MMTs reported in the literature. Additionally, a peak at 26.81 (2 theta) characteristic of quartz was observed.

According to the SEM (Figure 1d), the surface of MMT featured a typical flake-layered morphology [28]. In addition, the clay presented interesting values for total pore volume and average pore width in terms of the adsorption process. It showed a specific surface area (BET) (Table 1) higher than that reported for other natural clays such as Kaolin [29], beidellite and montmorillonites [30] from other regions.

According to the aforementioned results, the MMT from Colombia showed good characteristics of surface and chemical composition, which is promising for adsorption processes in water treatment. Thus, in the following sections, the MMT will be tested for the removal of both anionic and cationic dyes and pharmaceuticals present in water.

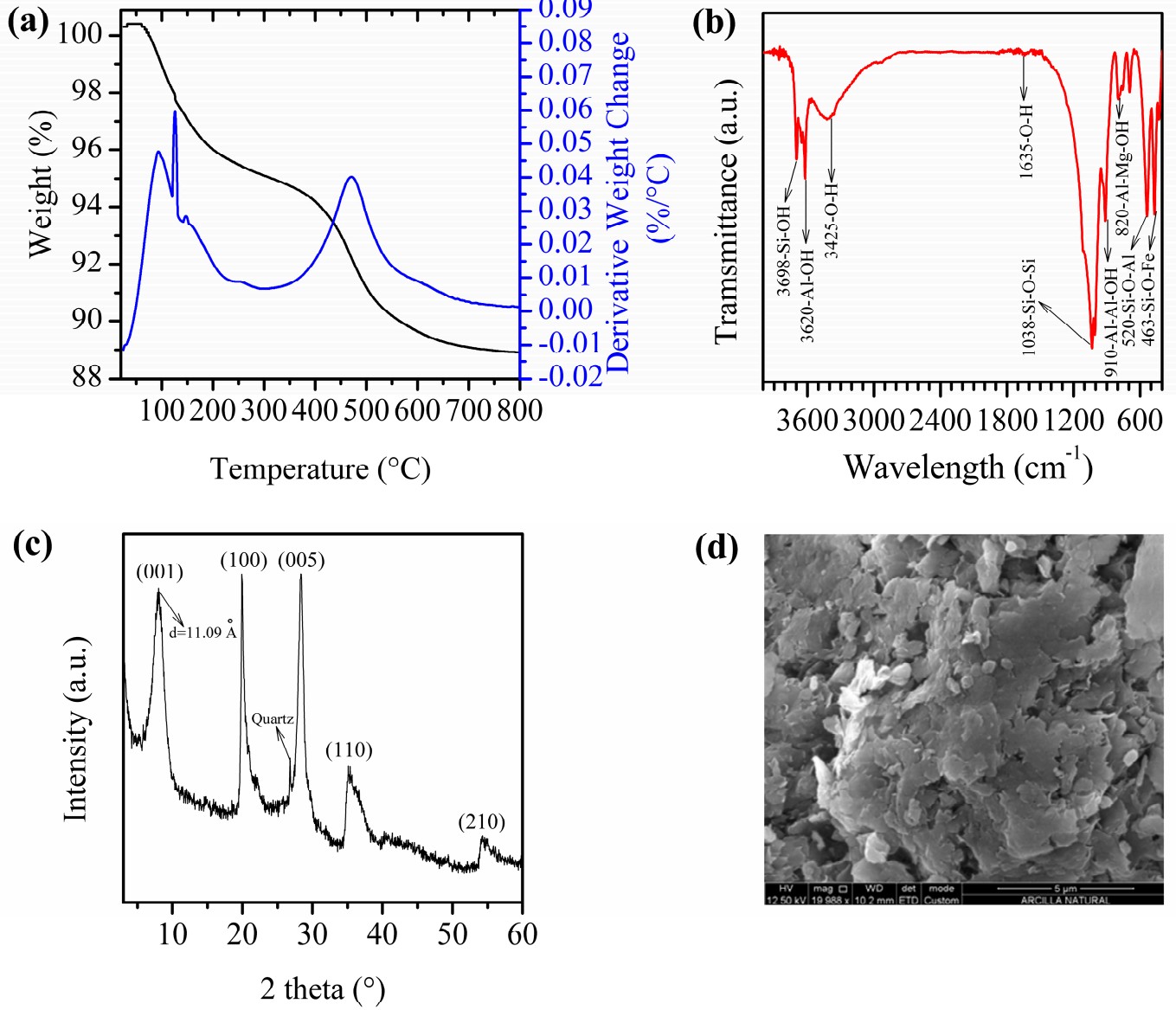

**Figure 1.** Physicochemical characterization of MMT. (**a**) TGA profile. Coordinates: Black color: Weight (%) vs. Temperature (°C); blue color: Derivative weight change (%/°C) vs. Temperature (°C); (**b**) FTIR spectra; (**c**) XRD analysis; (**d**) SEM micrograph.

**Table 1.** EDS analysis, specific surface area (BET) and PZC of MMT.

| EDS Analysis | | | |
|---|---|---|---|
| **Element** | **Wt (%)** | **Element** | **Wt (%)** |
| C | 3.25 | Mg | 2.19 |
| N | 0.76 | Al | 17.13 |
| O | 29.01 | Si | 36.50 |
| Na | 0.13 | Fe | 11.02 |
| **Nitrogen physisorption analysis** | | | |
| Specific surface area (BET) 82.5 m$^2$ g$^{-1}$ | | | |
| Total pore volume 0.004 cm$^3$ g$^{-1}$ | | | |
| Average pore width 65.5 nm | | | |
| **PZC** | | | |
| 2.6 ± 0.1 | | | |

### 3.2. Adsorption of Anionic and Cationic Dyes in Distilled Water Using MMT

The efficiency of MMT at absorbing both anionic (IC and CR) and cationic dyes (CV and MB) was evaluated, as shown in Figure 2. The results showed that MMT reached equilibrium in 60 min, following the order: CV > MB > CR > IC. The adsorption of the dyes using MMT can be attributed to the development of the surface area (BET), surface charge and the interlayer spacing of the natural clay [29]. The XRD analysis of MMT (Figure 1c) showed an inter-planar spacing of 11.09 Å. Therefore, due to the higher molecular size in terms of length, height or width of CV (11.37 Å × 8.70 Å × 10.25 Å) [31], MB (14.3 Å × 6.10 Å × 4.0 Å) [32], IC (17.71 Å × 8.03 Å × 6.16 Å) [33] and CR (26.2 Å × 7.40 Å × 4.30 Å) [32] none of the contaminants could enter the clay gaps. In this way, it can be assumed that there was no ion exchange between the pollutants and the Mg, Al, Si, Na and Fe ions present in the MMT as reported by the EDS analysis (Table 1). This suggests that the adsorption of the tested pollutants occurs via a different surface phenomenon.

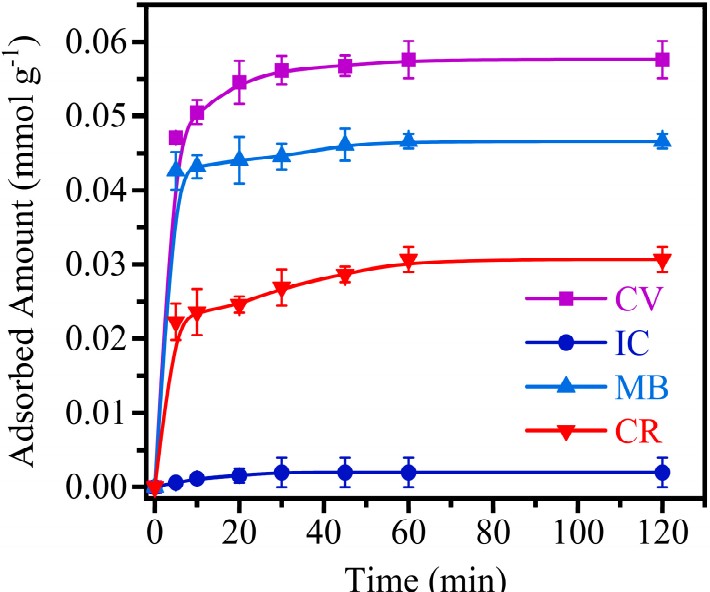

**Figure 2.** Amount of IC, CR, MB and CV in distilled water adsorbed onto MMT. Conditions: dye concentration $1.23 \times 10^{-2}$ mmol $L^{-1}$, adsorbent dose 0.2 g $L^{-1}$, particle size <200 μm, pH: IC: 5.7; CR: 5.7; MB: 5.6; CV: 6.0, temperature 25 °C, stirring rate 200 rpm.

The PZC of the adsorbent was calculated (Figure S1) and is reported in Table 1. Considering the pH of the CV solution (6.0) and its pKa value (9.39) [34], the dye was positively charged (Figure S2), thus producing favorable electrostatic attraction with the negative charges of the material. In the case of MB, despite it also being a cationic dye, the MMT showed lower removal compared to the CV. This is because MB has a pKa of 5.6 [35] (Figure S3), which is similar to the pH of the solution. Therefore, 50% of the pollutant is positively charged, presenting an attraction with the MMT, while the other 50% of the molecule is in its neutral form, which limits the amount of MB adsorbed.

On the other hand, the adsorbent did not show significant adsorption of IC (Figure 2). This is probably due to the fact that the dye is negatively charged [36] (Figure S4) and electronic repulsions take place. Surprisingly, a significant amount of the other anionic dye (CR) was adsorbed onto the MMT. At a pH of 5.7, CR with a pKa of 4.1 [37] is found in its negative form with some protonated species (Figure S5). Therefore, a fraction of protonated CR molecules can experience attraction due to the negatively charged surface of the clay.

The adsorption results obtained for the four dyes are noteworthy. Nevertheless, the adsorbed amount of these pollutants can change depending on the type of dye and the variation of the adsorbent dose. The effect of these parameters is discussed below.

### 3.3. Effects of MMT Dose and Dye Type on the Amount of Adsorbed Organic Pollutants

The effect of the adsorbent dose on the amount of IC, CR, MB and CV adsorbed onto MMT is reported in Figure 3. The Figure shows that for each of the experiments, the amount of adsorbed dye per unit mass of adsorbent decreases with an increase in adsorbent dose. This is due to the concentration gradient between solute concentrations in the solution and the adsorbent surface. Thus, with increasing adsorbent dosage, the amount of dye adsorbed by unit weight of adsorbent is reduced, causing a decrease in adsorption capacity. However, there is an increase in the removal percentage of IC (from 3.2 to 16%), CR (from 50 to 88%), MB (from 76 to 91%), and CV (from 90 to 99%), with an increase in MMT dosage from 0.2 to 1 g L$^{-1}$ (Figure S6). This can be attributed to the increased surface area and the availability of more adsorption sites [12]. However, at doses of MMT above 1 g L$^{-1}$ there is an excess of clay, and the adsorption of pollutants does not increase. Therefore, under the conditions studied, a dose of 1 g L$^{-1}$ will avoid incurring additional process costs.

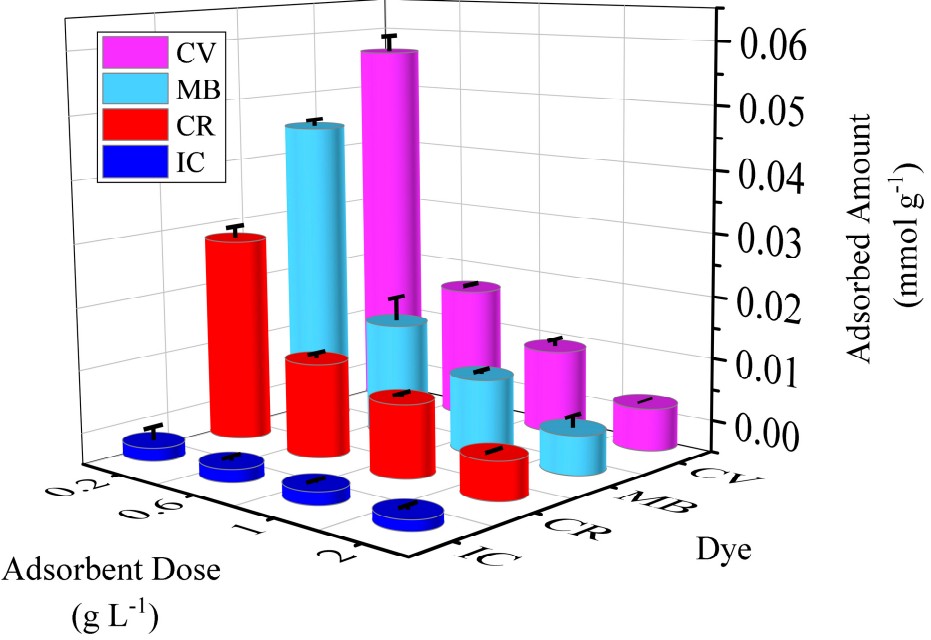

**Figure 3.** Effect of adsorbent dose on the adsorbed amount of IC, CR, MB and CV by MMT. Conditions: dye concentration 1.23 × 10$^{-2}$ mmol L$^{-1}$, adsorbent dose 0.2–2 g L$^{-1}$, particle size < 200 μm, pH: IC: 5.7; CR: 6.5; MB: 5.6; CV: 7.3, temperature 25 °C, time 60 min, stirring rate 200 rpm.

Moreover, Figure 3 shows the effect of the dye type, where the adsorption of the cationic dyes was greater than that experienced by the anionic dyes, following the order of CV > MB > CR > IC. This is due to the affinity for the charges analyzed, considering the PZC of the MMT, the pKa of the pollutant and the pH of the dyes in solution, as detailed in Section 3.2.

Consequently, the amounts of dye removed by the different doses of clay indicate the type of adsorption onto MMT experienced by IC, CR, MB, and CV. To evaluate this phenomenon, the adsorption isotherms of the process are analyzed in the next session.

### 3.4. Study of Adsorption Isotherms

Several adsorption isotherm models (Langmuir, Freundlich and Redlich–Peterson) were used to analyze the equilibrium data obtained in the present study (Figure S7). Mathematical adjustments were made for the Langmuir, Freundlich and Redlich–Peterson models during the IC, CR, MB, CV adsorption (Figures S8, S9, S10 and S11, respectively) and the parameters are shown in Table 2. According to the results of the Langmuir model, the dye adsorption presented affinity with the binding sites ($K_L$), and showed a maximum capacity, $q_m$, of 18.2, 16.2, 14.1 and 1.07 mg g$^{-1}$ for CV, CR, MB and IC,

respectively. However, the $q_m$ in mmol g$^{-1}$ indicated that adsorption onto MMT followed the order: CV > MB > CR > IC, thus confirming the affinity of MMT for positives charges. Additionally, a separation factor $R_L$ in a range between 0 and 1 was obtained for dye adsorption onto MMT, indicating that in all cases the adsorption process was favorable.

**Table 2.** Adsorption parameters for Langmuir, Freundlich and Redlich- Peterson isotherms for the removal of IC, CR, MB and CV from distilled water using MMT. Conditions: dye concentration $1.23 \times 10^{-5}$ mmol L$^{-1}$ (IC: 5.74 mg L$^{-1}$, CR: 8.57 mg L$^{-1}$, MB: 3.93 mg L$^{-1}$ and CV: 4.85 mg L$^{-1}$), adsorbent dose 0.2–2 g L$^{-1}$, particle size < 200 μm, pH: IC: 5.7; CR: 6.5; MB: 5.6; CV: 7.3, time 60 min, temperature 25 °C, stirring rate 200 rpm.

| Isotherm Models | Parameters | Dye | | | |
|---|---|---|---|---|---|
| | | IC | CR | MB | CV |
| Langmuir | $q_m$ (mg g$^{-1}$) | 1.07 | 16.2 | 14.1 | 18.2 |
| | $q_m$ (mmol g$^{-1}$) | $2.29 \times 10^{-3}$ | $2.33 \times 10^{-2}$ | $4.41 \times 10^{-2}$ | $4.62 \times 10^{-2}$ |
| | $K_L$ (L mg$^{-1}$) | 1.18 | 0.287 | 0.496 | 0.642 |
| | $K_L$ (L mmol$^{-1}$) | 550 | 200 | 159 | 253 |
| | $R_L$ | $4.23 \times 10^{-3}$– $4.25 \times 10^{-4}$ | $1.72 \times 10^{-2}$– $1.74 \times 10^{-3}$ | $9.98 \times 10^{-3}$– $1.01 \times 10^{-3}$ | $7.73 \times 10^{-3}$– $7.78 \times 10^{-4}$ |
| | R$^2$ | 0.9992 | 0.9909 | 0.9914 | 0.9999 |
| | APE (%) | 0.913 | 5.46 | 1.96 | 0.981 |
| | Δq (%) | 1.01 | 7.74 | 3.40 | 1.47 |
| Freundlich | $K_F$ (mg g$^{-1}$)(L mg$^{-1}$)$^{1/n}$ | 1.58 | 13.9 | 12.9 | 26.1 |
| | $n$ | 4.44 | 1.55 | 0.719 | 0.688 |
| | R$^2$ | 0.9885 | 0.9899 | 0.9870 | 0.9884 |
| | APE (%) | 3.51 | 7.44 | 4.70 | 5.21 |
| | Δq (%) | 5.43 | 11.48 | 6.34 | 7.37 |
| Redlich-Peterson | $K_{RP}$ (L g$^{-1}$) | 133 | 360 | 361 | 187 |
| | $a_R \left(\text{L mg}^{-1}\right)^{\beta}$ | 155 | 47.7 | 41.9 | 19.1 |
| | $\beta$ | 0.95 | 0.91 | 0.89 | 0.95 |
| | R$^2$ | 0.9990 | 0.9975 | 0.9997 | 0.9999 |
| | APE (%) | 0.715 | 0.825 | 0.426 | 0.261 |
| | Δq (%) | 0.984 | 1.35 | 0.640 | 0.554 |

Moreover, the adjustment of data to the Freundlich model is reported in Table 2, which shows adsorption capacity ($K_F$) values of 26.1, 13.9, 12.9 and 1.58 mg$^{1-1/n}$ L$^{1/n}$ g$^{-1}$ for CV, CR, MB and IC, respectively. Only CV presents a $n$ value in the range of 1–10, indicating favorable sorption for this pollutant.

However, the Langmuir isotherm model has a higher regression coefficient R$^2$ and a lower value of APE and Δq than the Freundlich model (Table 2), indicating that the Langmuir model has a better adjustment for the adsorption process of MMT with CV, CR, MB and IC. The results suggest monolayer adsorption for all the dyes onto the MMT surface.

Consequently, the Redlich-Peterson model was used to confirm the adjustment to the Langmuir model. The results of the Redlich-Peterson model are reported in Table 2. The model showed high values of R$^2$, and low values of APE and Δq for adsorption. Likewise, the values of $\beta$ tend to 1 in all the experiments with dyes, which confirmed the best fit of the data with the Langmuir model and suggests a monolayer-type adsorption.

Table 3 shows the $q_m$ values for the removal of IC, CR, MB and CV using MMT. In some cases, the clay presented a $q_m$ greater than that reported by other adsorbents. However, there are also materials that presented a $q_m$ greater than the value found in this investigation, possibly because these materials were rigorously washed with NaOH and/or subjected to high temperatures or doped with a metal, thus providing an improvement in the surface of the material, and consequently, an increase in its adsorbent capacity.

**Table 3.** Comparison of maximum adsorption capacity ($q_m$) of MMT for the removal of IC, CR, MB and CV against other adsorbents reported in the literature.

| Adsorbed Dye | Adsorbent | $q_m$ (mg g$^{-1}$) | $q_m$ (mmol g$^{-1}$) | Reference |
|---|---|---|---|---|
| IC | Abrasive spherical materials made of rice husk ash | 0.4 | $8.58 \times 10^{-4}$ | [17] |
| | MMT | 1.07 | $2.29 \times 10^{-3}$ | Present Study |
| | Fe-zeolitic | 32.8 | $7.04 \times 10^{-2}$ | [38] |
| | Activated carbon with ZnCl$_2$ from rice husk | 36.6 | $7.85 \times 10^{-2}$ | [18] |
| CR | Algerian kaolin | 5.94 | $8.53 \times 10^{-3}$ | [39] |
| | MMT | 16.2 | $2.33 \times 10^{-2}$ | Present Study |
| | Copolymer of luffa cylindrica fibre | 19.2 | $2.76 \times 10^{-2}$ | [40] |
| | Palygorskite clay | 51.2 | $7.35 \times 10^{-2}$ | [41] |
| MB | Sawdust | 7.84 | $2.45 \times 10^{-2}$ | [16] |
| | Kaolinite | 8.88 | $2.78 \times 10^{-2}$ | [42] |
| | MMT | 14.1 | $4.41 \times 10^{-2}$ | Present Study |
| | Mesoporous Iraqi red kaolin clay | 240 | $7.52 \times 10^{-1}$ | [43] |
| CV | Composite of typha latifolia activated carbon | 2.37 | $6.02 \times 10^{-3}$ | [44] |
| | Nano mesocellular foam silica | 6.60 | $1.68 \times 10^{-2}$ | [45] |
| | MMT | 18.2 | $4.62 \times 10^{-2}$ | Present Study |
| | Tunisian Smectite Clay | 86.5 | $2.20 \times 10^{-1}$ | [46] |

The MMT showed interesting $q_m$ values for CV, CR, MB and IC. However, the speed with which each of these processes occur is still unknown. Therefore, in the following section, the adsorption kinetics of MMT for each of the dyes will be studied.

### 3.5. Study of Adsorption Kinetics

The kinetics of the process were determined using different doses of MMT (Figure S7). The adsorption kinetics of IC, CR, MB and CV were then modeled with the pseudo-first order and pseudo-second order kinetics (Figures S12, S13, S14 and S15, respectively) and the results are shown in Table 4. The pseudo-first order and pseudo-second order models showed that q$_{eq, cal}$ is similar to q$_{eq, exp}$. However, the pseudo-second order model presented R$^2$ values >0.99 and percentages of *APE* and $\Delta q$ lower than those reported by the pseudo-first order model. This indicates that the pseudo-second order model is suitable to explain the results obtained. Thus, it is possible to assume that two active adsorption sites on MMT are able to adsorb one molecule of pollutant. In addition, Table 4 shows that increasing the initial MMT dose also increases the pseudo-second order kinetic rate constant, $k_2$. This confirms what was mentioned in Section 3.3, that at higher doses of clay, the active adsorption sites increase, and consequently, the equilibrium in the adsorption of all dyes is obtained more quickly.

The high correlation of the pseudo-second order kinetic equation with the kinetic data is consistent with previous results from the literature which deal with the adsorption of dyes onto clays, e.g., IC adsorption onto CdSe-Montmorillonite nanocomposites [47], CR adsorption onto carboxymethyl chitosan hybrid Montmorillonite [48], MB adsorption onto modified montmorillonite with magnetic nanoparticles and surfactants, and CV adsorption onto Na-Montmorillonite Nano Clay [49].

From the results obtained throughout this study, it can be inferred that even if MMT is able to remove negatively charged dyes, the material has a greater affinity, and is therefore more efficient, with positively charged dyes. In spite of this, the functional groups that participate in the adsorption process are unknown, and accordingly, this topic will be explored in next section.

**Table 4.** Kinetics of the removal of IC, CR, MB and CV from distilled water using MMT. Conditions: dye concentration $1.23 \times 10^{-2}$ mmol L$^{-1}$ (IC: 5.74 mg L$^{-1}$, CR: 8.57 mg L$^{-1}$, MB: 3.93 mg L$^{-1}$ and CV: 4.85 mg L$^{-1}$), adsorbent dose 0.2–2 g L$^{-1}$, particle size < 200 µm, pH: IC: 5.7; CR: 6.5; MB: 5.6; CV: 7.3, temperature 25 °C, time 60 min, stirring rate 200 rpm.

| | | | | | | | | | |
|---|---|---|---|---|---|---|---|---|---|
| **Pseudo-First Order Model** | | | | | | | | | |
| **Dye** | **Adsorbent Dose (g L$^{-1}$)** | **$q_{eq, exp}$ (mg g$^{-1}$)** | **$q_{eq, exp}$ (mmol g$^{-1}$)** | **$q_{eq, cal}$ (mg g$^{-1}$)** | **$q_{eq, cal}$ (mmol g$^{-1}$)** | **$k_1$ (min$^{-1}$)** | **$R^2$** | **$APE$ (%)** | **$\Delta q$ (%)** |
| IC | 0.2 | 0.919 | $1.97 \times 10^{-3}$ | 0.957 | $2.05 \times 10^{-3}$ | 0.0758 | 0.9912 | 6.05 | 9.61 |
| | 0.6 | 0.911 | $1.95 \times 10^{-3}$ | 0.942 | $2.02 \times 10^{-3}$ | 0.0407 | 0.9887 | 4.65 | 7.59 |
| | 1 | 0.909 | $1.95 \times 10^{-3}$ | 0.856 | $1.84 \times 10^{-3}$ | 0.146 | 0.9701 | 6.83 | 10.3 |
| | 2 | 0.722 | $1.55 \times 10^{-3}$ | 0.684 | $1.47 \times 10^{-3}$ | 0.160 | 0.9803 | 7.05 | 10.0 |
| CR | 0.2 | 21.4 | $3.07 \times 10^{-2}$ | 18.2 | $2.61 \times 10^{-2}$ | 6.01 | 0.9191 | 14.8 | 19.0 |
| | 0.6 | 9.81 | $1.41 \times 10^{-2}$ | 8.49 | $1.22 \times 10^{-2}$ | 17.8 | 0.9404 | 13.4 | 18.9 |
| | 1 | 7.56 | $1.09 \times 10^{-2}$ | 7.81 | $1.12 \times 10^{-2}$ | 0.209 | 0.9802 | 3.30 | 4.93 |
| | 2 | 4.06 | $5.83 \times 10^{-3}$ | 3.83 | $5.50 \times 10^{-3}$ | 7.75 | 0.9618 | 5.72 | 8.20 |
| MB | 0.2 | 14.9 | $4.66 \times 10^{-2}$ | 14.2 | $4.44 \times 10^{-2}$ | 4.15 | 0.9832 | 4.76 | 6.76 |
| | 0.6 | 5.50 | $1.72 \times 10^{-2}$ | 5.46 | $1.71 \times 10^{-2}$ | 0.809 | 0.9891 | 0.789 | 1.13 |
| | 1 | 3.59 | $1.12 \times 10^{-2}$ | 3.52 | $1.10 \times 10^{-2}$ | 18.8 | 0.9999 | 1.87 | 2.68 |
| | 2 | 1.87 | $5.85 \times 10^{-3}$ | 1.82 | $5.69 \times 10^{-3}$ | 4.14 | 0.9873 | 2.61 | 3.70 |
| CV | 0.2 | 22.7 | $5.76 \times 10^{-2}$ | 21.2 | $5.38 \times 10^{-2}$ | 18.3 | 0.9669 | 6.60 | 9.35 |
| | 0.6 | 7.87 | $2.00 \times 10^{-2}$ | 7.69 | $1.95 \times 10^{-2}$ | 0.371 | 0.9767 | 2.29 | 3.24 |
| | 1 | 4.96 | $1.26 \times 10^{-2}$ | 4.89 | $1.24 \times 10^{-2}$ | 17.96 | 0.9864 | 1.35 | 1.92 |
| | 2 | 2.51 | $6.37 \times 10^{-3}$ | 2.50 | $6.35 \times 10^{-3}$ | 0.741 | 0.9877 | 0.398 | 0.563 |

**Table 4.** *Cont.*

| | | | | | | | | | | |
|---|---|---|---|---|---|---|---|---|---|---|
| **Pseudo-Second Order Model** | | | | | | | | | | |
| **Dye** | **Adsorbent Dose (g L$^{-1}$)** | **$q_{eq, exp}$ (mg g$^{-1}$)** | **$q_{eq, exp}$ (mmol g$^{-1}$)** | **$q_{eq, cal}$ (mg g$^{-1}$)** | **$q_{eq, cal}$ (mmol g$^{-1}$)** | **$k_2$ (mg g$^{-1}$ min$^{-1}$)** | **$k_2$ (mmol g$^{-1}$ min$^{-1}$)** | **$R^2$** | **$APE$ (%)** | **$\Delta q$ (%)** |
| IC | 0.2 | 0.919 | $1.97 \times 10^{-3}$ | 0.983 | $2.11 \times 10^{-3}$ | 0.0636 | $1.36 \times 10^{-4}$ | 0.9967 | 3.50 | 6.67 |
| | 0.6 | 0.911 | $1.95 \times 10^{-3}$ | 0.934 | $2.00 \times 10^{-3}$ | 0.0674 | $1.45 \times 10^{-4}$ | 0.9981 | 4.62 | 7.01 |
| | 1 | 0.909 | $1.95 \times 10^{-3}$ | 0.906 | $1.94 \times 10^{-3}$ | 0.207 | $4.44 \times 10^{-4}$ | 0.9919 | 2.59 | 4.16 |
| | 2 | 0.722 | $1.55 \times 10^{-3}$ | 0.767 | $1.64 \times 10^{-3}$ | 0.289 | $6.20 \times 10^{-4}$ | 0.9963 | 6.47 | 9.1 |
| CR | 0.2 | 21.4 | $3.07 \times 10^{-2}$ | 20.6 | $2.96 \times 10^{-2}$ | 0.0239 | $3.43 \times 10^{-5}$ | 0.9901 | 3.61 | 5.36 |
| | 0.6 | 9.81 | $1.41 \times 10^{-2}$ | 9.41 | $1.35 \times 10^{-2}$ | 0.0634 | $9.10 \times 10^{-5}$ | 0.9974 | 3.99 | 5.89 |
| | 1 | 7.56 | $1.09 \times 10^{-2}$ | 7.66 | $1.10 \times 10^{-2}$ | 0.0815 | $1.17 \times 10^{-4}$ | 0.9956 | 1.31 | 2.43 |
| | 2 | 4.06 | $5.83 \times 10^{-3}$ | 4.15 | $5.96 \times 10^{-3}$ | 0.180 | $2.58 \times 10^{-4}$ | 0.9968 | 2.15 | 3.35 |
| MB | 0.2 | 14.9 | $4.66 \times 10^{-2}$ | 14.7 | $4.60 \times 10^{-2}$ | 0.147 | $4.60 \times 10^{-4}$ | 0.9981 | 1.41 | 2.08 |
| | 0.6 | 5.50 | $1.72 \times 10^{-2}$ | 5.49 | $1.72 \times 10^{-2}$ | 1.39 | $4.35 \times 10^{-3}$ | 0.9999 | 0.244 | 0.403 |
| | 1 | 3.59 | $1.12 \times 10^{-2}$ | 3.52 | $1.10 \times 10^{-2}$ | 1.76 | $5.50 \times 10^{-3}$ | 0.9999 | 1.87 | 2.68 |
| | 2 | 1.87 | $5.85 \times 10^{-3}$ | 1.85 | $5.78 \times 10^{-3}$ | 1.80 | $5.63 \times 10^{-3}$ | 0.9999 | 1 | 1.45 |
| CV | 0.2 | 22.7 | $5.76 \times 10^{-2}$ | 21.2 | $5.38 \times 10^{-2}$ | 0.0515 | $1.31 \times 10^{-4}$ | 0.9973 | 6.60 | 9.35 |
| | 0.6 | 7.87 | $2.00 \times 10^{-2}$ | 7.99 | $2.03 \times 10^{-2}$ | 0.117 | $2.97 \times 10^{-4}$ | 0.9999 | 1.52 | 2.16 |
| | 1 | 4.96 | $1.26 \times 10^{-2}$ | 4.95 | $1.26 \times 10^{-2}$ | 1.28 | $3.25 \times 10^{-3}$ | 0.9999 | 0.145 | 0.223 |
| | 2 | 2.51 | $6.37 \times 10^{-3}$ | 2.52 | $6.40 \times 10^{-3}$ | 2.27 | $5.76 \times 10^{-3}$ | 0.9999 | 0.398 | 0.563 |

### 3.6. Identification of the Adsorption Sites on MMT for Removal of Dyes

Figure 4 shows the FTIR spectrum of MMT before and after the adsorption of each dye. For the case of IC (Figure 4a), after the removal of the pollutant, the FTIR spectrum of the clay shows changes in the bands at 3425 and 1635 cm$^{-1}$. These changes suggest that the adsorption reported for IC takes place via hydrogen bonds formed between water present in the clay and the hydrogen of the amine (Figure 5a). Moreover, the water on the MMT can form hydrogen bonds with the oxygens of the sulfonate group or with those of the cyclopentane in the dye (Figure 5a).

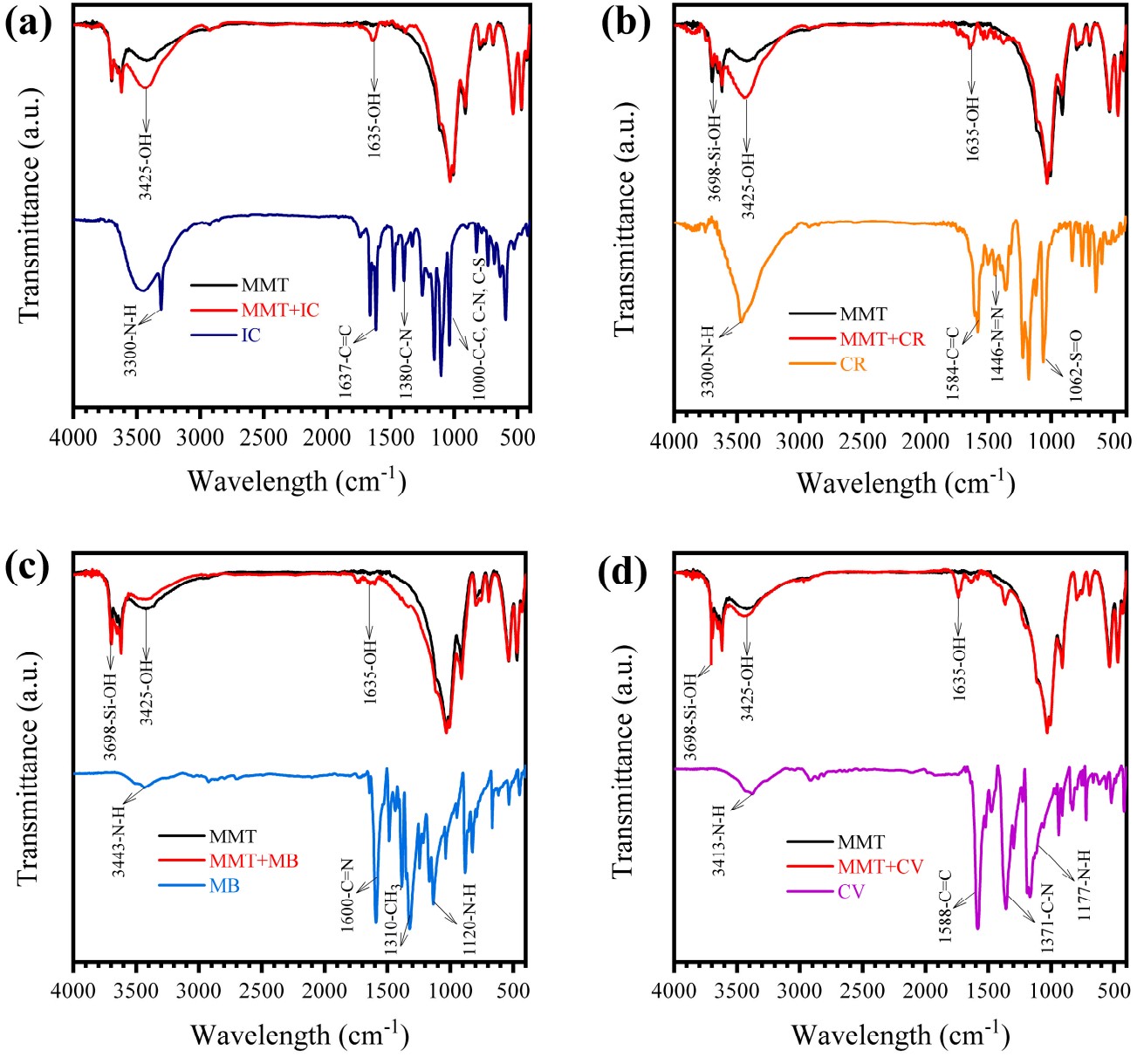

**Figure 4.** FTIR spectrum before and after the adsorption of the dyes in distilled water using MMT. (**a**) IC; (**b**) CR; (**c**) MB and (**d**) CV. Conditions: dye concentration $1.23 \times 10^{-5}$ mol L$^{-1}$, adsorbent dose 1 g L$^{-1}$, particle size <200 µm. pH: IC: 5.7; CR: 6.5; MB: 5.6; CV: 7.3, temperature 25 °C, stirring rate 200 rpm.

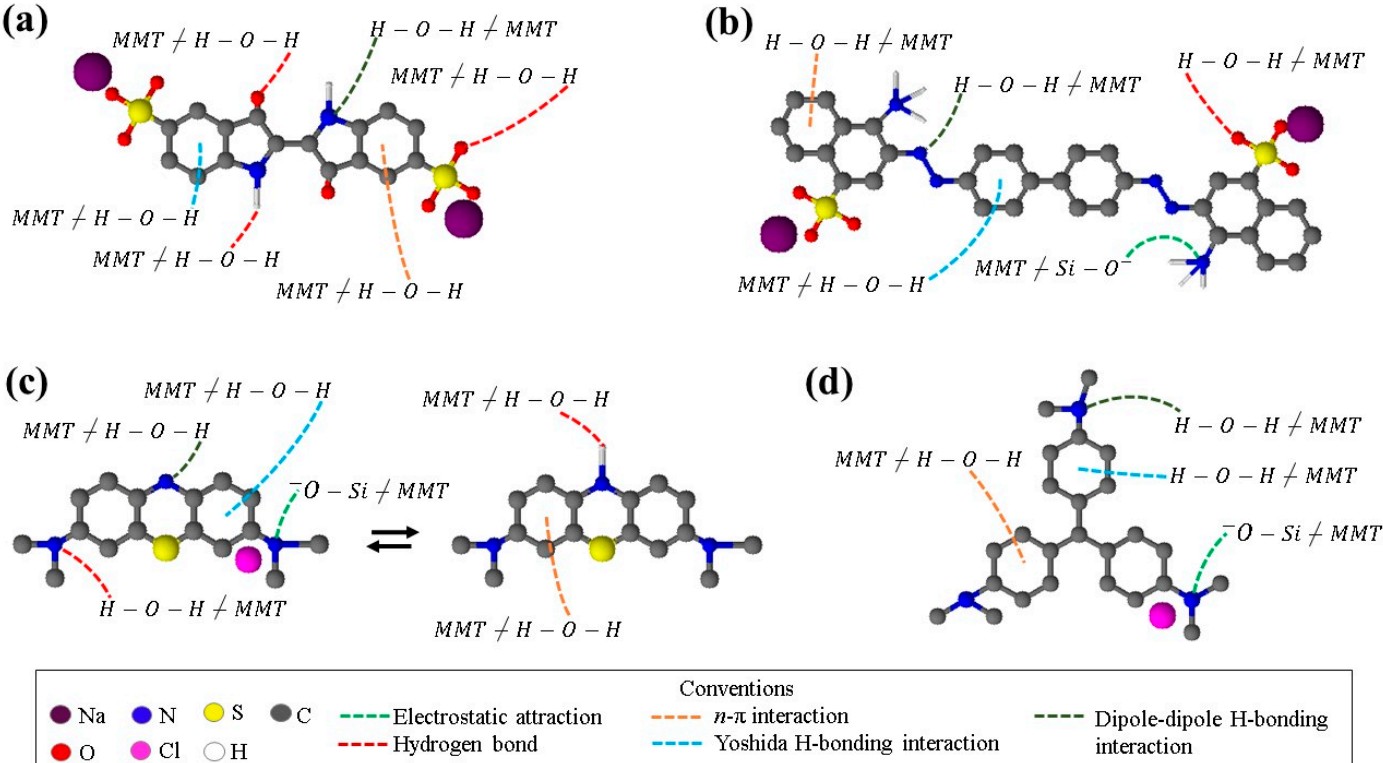

**Figure 5.** Most favorable interactions between MMT and the dyes during the adsorption process. (**a**) IC; (**b**) CR; (**c**) MB; (**d**) CV.

Figure 4b–d shows the FTIR spectra for CR, MB and CV, respectively, and for MMT before and after the adsorption process. For the three spectra of MMT after the adsorption process, changes are observed in the bands of the Si-OH groups at 3698 cm$^{-1}$. Similar to what was found for IC, there are variations in the bands at 3425 and 1635 cm$^{-1}$ of the water present in the MMT. These changes allow the most probable adsorption sites for each of the three dyes (CR, MB and CV) to be proposed.

The most probable interaction sites between MMT and CR removal are shown in Figure 5b. Thus, CR can be adsorbed by electrostatic attractions between the positively charged amine present in its aromatic ring and the Si-O$^-$ group of the MMT, which is deprotonated at the pH of the experiment (pKa of 4.8) [50]. In addition, there may be hydrogen bonds between the oxygens of the sulfonate group of CR and H within the interlaminar water of the clay.

For MB, Figure 5c illustrates electrostatic attractions between the positively charged amine of the dye and the Si-O$^-$ group of the MMT. Even hydrogen bonds can occur between the N of the neutral amine and the H of the water in the clay. Furthermore, MB can be in its neutral form, thus it can form hydrogen bonds between the hydrogen of the amine of the aromatic ring and the water present in the MMT.

Figure 5d shows the CV adsorption, which may occur by electrostatic attractions between the Si-O$^-$ of the clay and the positively charged amine of the dye.

Finally, Figure 5a–d also shows additional favorable interactions between MMT and the dyes (CV, MB, CR and IC, respectively). In this way, the aromatic rings of the contaminants with O or H of the water in the clay can form n-π interactions and Yoshida H-bonding interactions, respectively. Other possible interactions such as dipole-dipole H-bonding may also occur between the N of the dyes and H in the MMT.

### 3.7. Cost and Reuse of MMT

To further evaluate the feasibility of the technology, an economic analysis of the performance of the system was carried out and the results are shown in Table 5. A low

cost was reported for the production of MMT (5.76 USD kg$^{-1}$), which is associated with its abundance, easy acquisition and simple preparation process. The cost of using MMT to remove IC, CR, MB and CV was then calculated. The results indicate that MMT can remove more grams of CV than CR, MB and IC per USD. Therefore, MMT can be considered a low-cost and efficient alternative for CV removal.

**Table 5.** Cost of using MMT for the removal of dyes.

| Cost Production of MMT | | | |
|---|---|---|---|
| **Items** | **Unit Price** | **Consumption** | **Cost (USD)** |
| Power | 0.160 USD kWh$^{-1}$ | 21.2 kW h$^{-1}$ | 3.39 |
| MMT | 2.13 USD kg$^{-1}$ | 1.11 kg | 2.36 |
| Water | 0.950 USD m$^{-3}$ | 0.0110 m$^3$ | 0.0105 |
| Total Cost production MMT (USD kg$^{-1}$) | | | 5.76 |
| **Cost of MMT for removal of dyes** | | | |
| Dye | IC | CR | MB | CV |
| $q_m$ (g kg$^{-1}$) | 0.922 | 16.2 | 14.1 | 18.2 |
| (g dye/USD) | 0.160 | 2.81 | 2.45 | 3.16 |

The positive results obtained for removing CV using MMT are a good reason to study other properties of the material, such as its reusability. Therefore, the reuse of MMT after CV removal was explored. Figure 6 shows that the adsorption percentage of the dye decreases when the MMT is reused. However, clay can be effective for up to 4 cycles, reaching 76% of pollutant removal in the last cycle. This additional advantage of MMT favors its use in complex water treatment.

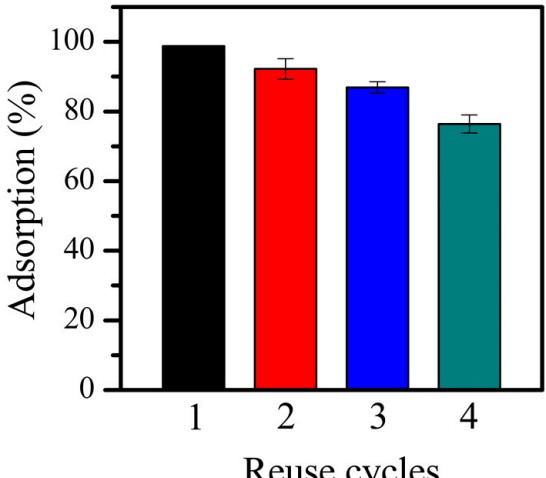

**Figure 6.** Reuse of MMT to remove CV from distilled water after 60 min. Conditions: CV: 4.85 mg L$^{-1}$, adsorbent dose 1 g L$^{-1}$, particle size < 200 μm, temperature 25 °C, stirring rate 200 rpm and pH of 7.3.

*3.8. Complex Matrix Effect*

The success of MMT in this study at removing organic contaminants present in complex waters would help persuade the industrial sector to use this material on a real scale. To further investigate this, the adsorption of CV (the dye with the best adsorption properties in this study) onto MMT in synthetic textile wastewater was explored. Considering the high levels of pharmaceutical contamination and that fact that urine represents a primary source of wastewater contamination, the ability of MMT to remove an antibiotic and an analgesic in simulated urine was also studied.

### 3.8.1. CV Adsorption in Textile Wastewater

Figure 7a shows the use of MMT to remove CV from both distilled water and textile wastewater. In distilled water, it is possible to remove practically 100% of the dye in less than 60 min because at the study pH (7.3) the contaminant is favored by electrostatic attractions and hydrogen bonds, as demonstrated in Section 3.6. However, in textile wastewater the adsorption decreased by around 14%, possibly due to the increase in pH (value of 10). Thus, CV was in its neutral form (Figure S2) and the pollutant adsorption took place via hydrogen bonds that formed between the N groups of the dye and the water in the interlayer spacing of the MMT. Furthermore, this small detrimental matrix effect may be associated with the components present in the textile wastewater [14]. For example, starch is an organic polymer with a chemical structure rich in H and O. It can therefore also adsorb onto the clay via hydrogen bonds. Consequently, competition occurs with the MMT, thus reducing the removal efficiency of the dye in this complex matrix.

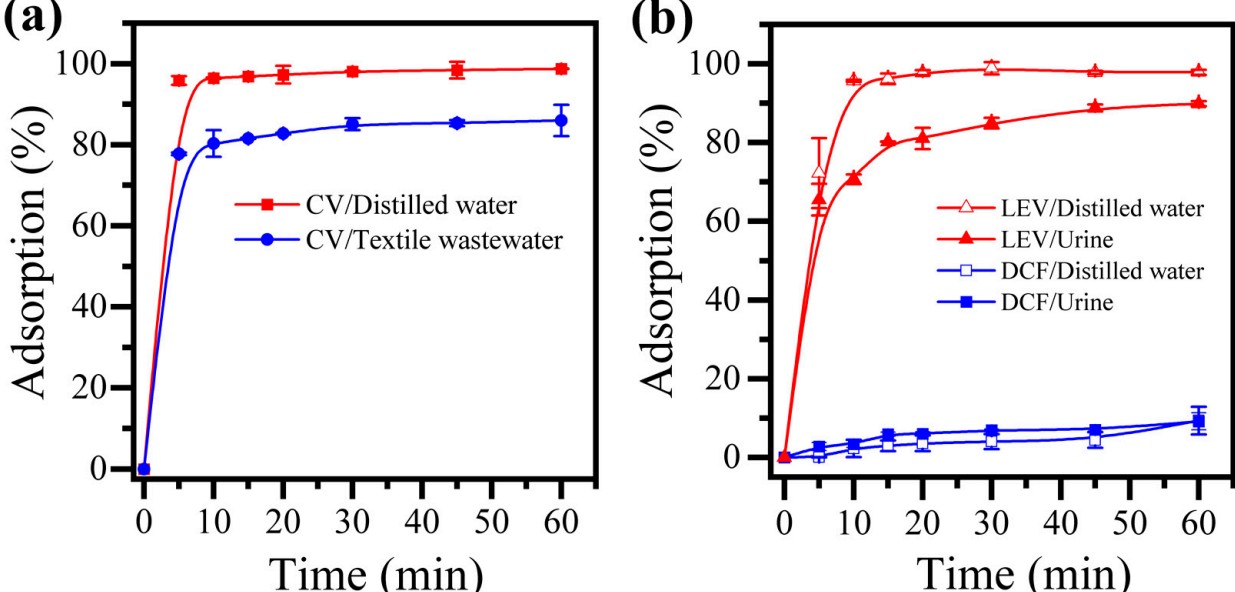

**Figure 7.** Complex matrix effect using MMT. (**a**) Adsorption of CV in distilled water and textile wastewater; (**b**) LEV and DCF adsorption in distilled water and urine. Conditions: pollutant concentration $1.23 \times 10^{-2}$ mmol L$^{-1}$ (CV: 4.85 mg L$^{-1}$, LEV: 4.44 mg L$^{-1}$, DCF: 3.64 mg L$^{-1}$), adsorbent dose 1 g L$^{-1}$, particle size <200 μm, temperature 25 °C, stirring rate 200 rpm. Experiments were performed at natural pH, for example 7.3 for CV in distilled water and 10 for CV in textile wastewater. For LEV and DCF in distilled water, the pH was 6.3 and 6.0 respectively, while in urine the pH was 6.0 for both pharmaceuticals.

### 3.8.2. Pharmaceutical Adsorption in Urine

The efficiency of MMT was evaluated for the adsorption of LEV and DCF present in urine and the results obtained are shown in Figure 7b. Even after 60 min, the adsorption of DCF in distilled water and in urine is less than 10%. DCF has a pKa of 4.15 [51]. Therefore, at the pH of the solution, in both matrices the pollutant will have most of the molecules negatively charged causing an electrostatic repulsion within the surface of the clay, which is negatively charged due to its low PZC value (Table 1). In contrast, a high and efficient removal of LEV was observed in distilled water after only 5 min (95%). The LEV had pKa values of 6.0 and 8.1 [52]. Therefore, at the experimental pH, LEV is in its zwitterionic form, which enables electrostatic attractions between its positive charge and the negatively charged surface of the material. Additionally, LEV adsorption in urine showed a decrease of 8%, possibly because this type of matrix is characterized by a high content of inorganic salts rich in ions [53] such as Na$^+$, K$^+$, Ca$^{2+}$ and Mg$^{2+}$ that can compete with the pollutant for the active sites present in the MMT. Thus, these results show the affinity of MMT in the

removal of pharmaceuticals with a positive charge, even in a highly complex matrix such as urine.

## 4. Conclusions

This study analyzed the potential of MMT, a natural clay from Colombia, to remove dyes and pharmaceuticals from wastewater. The MMT presented a high content of hydroxyl groups and bounded water. Additionally, it had a high BET surface area (82.5 $m^2\,g^{-1}$) and wide interlaminar spacing (11.09 Å), meaning that it had greater adsorption characteristics than other Montmorillonite clays.

At the natural pH of the experiments, MMT efficiently removed pollutants that had positive charges, such as the CV and MB dyes and the levofloxacin antibiotic. This was due to the favorable attraction between the positive charges of the pollutants and the negative surface of the clay (PZC 2.6).

The evaluation of the adsorbent dose showed that the increase in the adsorbent concentration increased the MMT sites and improved the removal of the pollutants. Furthermore, the equilibrium data for the adsorption of IC, CR, MB and CV onto MMT adjusted best to the Langmuir model, indicating that the pollutants form a monolayer on the surface of the clay. It is worth noting that the $q_m$ for CV elimination (18.2 mg $g^{-1}$) was even higher than that found for other materials such as composites of typha latifolia activated carbon (2.37 mg $g^{-1}$) and nano mesocellular foam silica (6.60 mg $g^{-1}$).

In addition, the adsorption kinetics fitted well with the pseudo-second order kinetic model, indicating that one pollutant molecule is adsorbed onto two active adsorption sites of MMT.

The removal of CV from textile wastewater using MMT was negatively affected to some extent, probably due to the presence of starch, while LEV adsorption in urine decreased slightly, maybe due to the presence of cations in the matrix. In addition, DCF removal was not possible due to the anionic nature of this pharmaceutical. According to the aforementioned results, MMT proved to be an effective and low-cost natural adsorbent with good reuse characteristics for the efficient removal of cationic organic pollutants, which suggests it is suitable for full-scale applications.

**Supplementary Materials:** The following supporting information can be downloaded at: https://www.mdpi.com/article/10.3390/w15061046/s1, Table S1: Chemical composition of fresh urine and textile wastewater; Figure S1: Determination of PZC value of MMT; Figure S2: CV Structure; Figure S3: MB Structure; Figure S4: IC Structure; Figure S5: CR Structure; Figure S6: Effect of MMT dose on the adsorption percentage of CV, MB, CR and IC after 60 min; Figure S7: IC, CR, MB and CV removal in distilled water using different doses of MMT; Figure S8: Adsorption isotherms for IC removal in distilled water using MMT as an adsorbent. (a) Langmuir model; (b) Freundlich model; (c) Redlich-Peterson model; Figure S9: Adsorption isotherms for CR removal in distilled water using MMT as an adsorbent. (a) Langmuir model; (b) Freundlich model; (c) Redlich-Peterson model; Figure S10: Adsorption isotherms for MB removal in distilled water using MMT as an adsorbent. (a) Langmuir model; (b) Freundlich model; (c) Redlich-Peterson model; Figure S11: Adsorption isotherms for CV removal in distilled water using MMT as an adsorbent. (a) Langmuir model; (b) Freundlich model; (c) Redlich-Peterson model; Figure S12: Kinetics of pseudo-first order model and pseudo-second order model for IC removal in distilled water using MMT as an adsorbent. (a) 0.2 g $L^{-1}$; (b) 0.6 g $L^{-1}$; (c) 1 g $L^{-1}$; (d) 2 g $L^{-1}$; Figure S13: Kinetics of pseudo-first order model and pseudo-second order model for CR removal in distilled water using MMT as an adsorbent. (a) 0.2 g $L^{-1}$; (b) 0.6 g $L^{-1}$; (c) 1 g $L^{-1}$; (d) 2 g $L^{-1}$; Figure S14: Kinetics of pseudo-first order model and pseudo-second order model for MB removal in distilled water using MMT as an adsorbent. (a) 0.2 g $L^{-1}$; (b) 0.6 g $L^{-1}$; (c) 1 g $L^{-1}$; (d) 2 g $L^{-1}$; Figure S15: Kinetics of pseudo-first order model and pseudo-second order model for CV removal in distilled water using MMT as an adsorbent. (a) 0.2 g $L^{-1}$; (b) 0.6 g $L^{-1}$; (c) 1 g $L^{-1}$; (d) 2 g $L^{-1}$.

**Author Contributions:** Conceptualization, M.P.-L., D.F.M. and R.A.T.-P.; methodology, M.P.-L. and D.F.M.; validation, M.P.-L.; formal analysis, M.P.-L.; investigation, M.P.-L.; resources, D.F.M. and R.A.T.-P.; writing—original draft preparation, M.P.-L.; writing—review and editing, M.P.-L., D.F.M.

and R.A.T.-P.; visualization, M.P.-L., D.F.M. and R.A.T.-P.; supervision, R.A.T.-P.; project administration, M.P.-L. and R.A.T.-P.; funding acquisition, R.A.T.-P. All authors have read and agreed to the published version of the manuscript.

**Funding:** This research received no external funding.

**Data Availability Statement:** Not applicable.

**Acknowledgments:** The authors wish to thank MINCIENCIAS for project No. 1115-852-69594 "Program to improve the quality of irrigation water through the elimination of emerging contaminants by advanced oxidation processes (PRO-CEC-AGUA)". M. Paredes-Laverde would like to thank the Gobernación del Departamento del Huila and MINCIENCIAS for her scholarship through the program "Becas de Excelencia Doctoral del Bicentenario—Primer Corte".

**Conflicts of Interest:** The authors declare no conflict of interest.

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
