# Peer review of "Montmorillonite-Based Natural Adsorbent from Colombia for the Removal of Organic Pollutants from Water: Isotherms, Kinetics, Nature of Pollutants, and Matrix Effects"

_water, doi:10.3390/w15061046_

Round 1
Reviewer 1 Report
1. It is suggested that the author add the research background, objectives and methods in the abstract and please add a sentence to explain the necessity of research.
2. Line 51 and Line 134. Eliminate multiple references. After that please check the manuscript thoroughly and eliminate all the lumps in the manuscript. This should be done by characterizing each reference individually. This can be done by mentioning 1 or 2 phrases per reference to show how it is different from the others and why it deserves mentioning.
3. In the conclusions, in addition to summarizing the actions taken and results, please strengthen the explanation of their significance. It is recommended to use quantitative reasoning comparing with appropriate benchmarks, especially those stemming from previous work.
4. It is suggested to provide more detailed experiment methods in Section 2.4.
Reviewer 2 Report
The manuscript deserves publication, however, major revisions are needed:
1. Please, strongly justify in the abstract and in the introduction section the novelty and originality of his/her work.
2. Please, describe in detail all experimental procedures used in order to allow their reproducibility.
3. Please, check the standard conditions of all figures, quality, dimensions, colors, descriptions and definition.
4. Please, revise all information of each one of the references used, including the style and format. Several of them do not follow the publication guidelines of the journal.
5. The selection of dyes should be justified.
6. Include information about the environmental levels in Colombia and international agencies in order to compare and evaluate the proposal here.
7. Discuss the kind of elimination of dyes by adsorption, and not by degradation or complete mineralization ways.
8. Economical costs should be reported.
10. Units format should be homogenized
11. Data and mathematical treatment of data to obtain the Table 2 and 4, should be reported as supplementary material. R2 is some cases is too high, but it is necessary to evaluate the linearity or non-linearity in the data and plots obtained.
12. kinetic behavior is evaluated from plots obtained. Please, report all plots obtained.
13. What is the reliability of the data? several data reports a non-uniform number of digits after the dot, but in some cases, no representative digits have been reported. The statistical treatment is not reliable.
14. Complex matrix effect is confuse and it is not complete, and also it lacks in results discussion.
15. several references are out of the scope of the manuscript. Revise
16. re-sue of the adsorption material should be reported.
17. What happens with the adsorption material is not used? What are the alternatives to treat it or release it?
Round 2
Reviewer 2 Report
All concerns have been carefully addressed by the authors and the revised version was significantly improved.
Author Response
Thank you for your pertinent contributions that helped us improve our paper.